# Geometric Descent Method for Convex Composite Minimization

**Shixiang Chen[1], Shiqian Ma[2], and Wei Liu[3]**

[1]Department of SEEM, The Chinese University of Hong Kong, Hong Kong
[2]Department of Mathematics, UC Davis, USA
[3]Tencent AI Lab, China

## Abstract

In this paper, we extend the geometric descent method recently proposed by Bubeck, Lee and Singh [1] to tackle nonsmooth and strongly convex composite problems. We prove that our proposed algorithm, dubbed geometric proximal gradient method (GeoPG), converges with a linear rate $(1 - 1/\sqrt{\kappa})$ and thus achieves the optimal rate among first-order methods, where $\kappa$ is the condition number of the problem. Numerical results on linear regression and logistic regression with elastic net regularization show that GeoPG compares favorably with Nesterov's accelerated proximal gradient method, especially when the problem is ill-conditioned.

## 1 Introduction

Recently, Bubeck, Lee and Singh proposed a geometric descent method (GeoD) for minimizing a smooth and strongly convex function [1]. They showed that GeoD achieves the same optimal rate as Nesterov's accelerated gradient method (AGM) [2, 3]. In this paper, we provide an extension of GeoD that minimizes a nonsmooth function in the composite form:

$$\min_{x \in \mathbb{R}^n} F(x) := f(x) + h(x), \tag{1.1}$$

where $f$ is $\alpha$-strongly convex and $\beta$-smooth (*i.e.*, $\nabla f$ is Lipschitz continuous with Lipschitz constant $\beta$), and $h$ is a closed nonsmooth convex function with simple proximal mapping. Commonly seen examples of $h$ include $\ell_1$ norm, $\ell_2$ norm, nuclear norm, and so on.

If $h$ vanishes, then the objective function of (1.1) becomes smooth and strongly convex. In this case, it is known that AGM converges with a linear rate $(1 - 1/\sqrt{\kappa})$, which is optimal among all first-order methods, where $\kappa = \beta/\alpha$ is the condition number of the problem. However, AGM lacks a clear geometric intuition, making it difficult to interpret. Recently, there has been much work on attempting to explain AGM or designing new algorithms with the same optimal rate (see, [4, 5, 1, 6, 7]). In particular, the GeoD method proposed in [1] has a clear geometric intuition that is in the flavor of the ellipsoid method [8]. The follow-up work [9, 10] attempted to improve the performance of GeoD by exploiting the gradient information from the past with a "limited-memory" idea. Moreover, Drusvyatskiy, Fazel and Roy [10] showed how to extend the suboptimal version of GeoD (with the convergence rate $(1 - 1/\kappa)$) to solve the composite problem (1.1). However, it was not clear how to extend the optimal version of GeoD to address (1.1), and the authors posed this as an open question. In this paper, we settle this question by proposing a geometric proximal gradient (GeoPG) algorithm which can solve the composite problem (1.1). We further show how to incorporate various techniques to improve the performance of the proposed algorithm.

**Notation.** We use $B(c, r^2) = \{x | \|x - c\|^2 \le r^2\}$ to denote the ball with center $c$ and radius $r$. We use $\text{Line}(x, y)$ to denote the line that connects $x$ and $y$, *i.e.*, $\{x + s(y - x), s \in \mathbb{R}\}$. For fixed $t \in (0, 1/\beta]$, we denote $x^+ := \text{Prox}_{th}(x - t\nabla f(x))$, where the proximal mapping $\text{Prox}_h(\cdot)$ is

defined as $\text{Prox}_h(x) := \text{argmin}_z \ h(z) + \frac{1}{2}\|z - x\|^2$. The proximal gradient of $F$ is defined as $G_t(x) := (x - x^+)/t$. It should be noted that $x^+ = x - tG_t(x)$. We also denote $x^{++} := x - G_t(x)/\alpha$. Note that both $x^+$ and $x^{++}$ are related to $t$, and we omit $t$ whenever there is no ambiguity.

The rest of this paper is organized as follows. In Section 2, we briefly review the GeoD method for solving smooth and strongly convex problems. In Section 3, we provide our GeoPG algorithm for solving nonsmooth problem (1.1) and analyze its convergence rate. We address two practical issues of the proposed method in Section 4, and incorporate two techniques: backtracking and limited memory, to cope with these issues. In Section 5, we report some numerical results of comparing GeoPG with Nesterov's accelerated proximal gradient method in solving linear regression and logistic regression problems with elastic net regularization. Finally, we conclude the paper in Section 6.

## 2 Geometric Descent Method for Smooth Problems

The GeoD method [1] solves (1.1) when $h \equiv 0$, in which the problem reduces to a smooth and strongly convex problem $\min \ f(x)$. We denote its optimal solution and optimal value as $x^*$ and $f^*$, respectively. Throughout this section, we fix $t = 1/\beta$, which together with $h \equiv 0$ implies that $x^+ = x - \nabla f(x)/\beta$ and $x^{++} = x - \nabla f(x)/\alpha$. We first briefly describe the basic idea of the suboptimal GeoD. Since $f$ is $\alpha$-strongly convex, the following inequality holds

$$f(x) + \langle \nabla f(x), y - x \rangle + \frac{\alpha}{2}\|y - x\|^2 \leq f(y), \ \forall x, y \in \mathbb{R}^n. \tag{2.1}$$

By letting $y = x^*$ in (2.1), it is easy to obtain

$$x^* \in B\big(x^{++}, \|\nabla f(x)\|^2/\alpha^2 - 2(f(x) - f^*)/\alpha\big), \forall x \in \mathbb{R}^n. \tag{2.2}$$

Note that the $\beta$-smoothness of $f$ implies

$$f(x^+) \leq f(x) - \|\nabla f(x)\|^2/(2\beta), \forall x \in \mathbb{R}^n. \tag{2.3}$$

Combining (2.2) and (2.3) yields $x^* \in B\big(x^{++}, (1 - 1/\kappa)\|\nabla f(x)\|^2/\alpha^2 - 2(f(x^+) - f^*)/\alpha\big)$. As a result, suppose that initially we have a ball $B(x_0, R_0^2)$ that contains $x^*$, then it follows that

$$x^* \in B\big(x_0, R_0^2\big) \cap B\big(x_0^{++}, (1 - 1/\kappa)\|\nabla f(x_0)\|^2/\alpha^2 - 2(f(x_0^+) - f^*)/\alpha\big). \tag{2.4}$$

Some simple algebraic calculations show that the squared radius of the minimum enclosing ball of the right hand side of (2.4) is no larger than $R_0^2(1 - 1/\kappa)$, *i.e.*, there exists some $x_1 \in \mathbb{R}^n$ such that $x^* \in B\big(x_1, R_0^2(1 - 1/\kappa)\big)$. Therefore, the squared radius of the initial ball shrinks by a factor $(1 - 1/\kappa)$. Repeating this process yields a linear convergent sequence $\{x_k\}$ with the convergence rate $(1 - 1/\kappa)$: $\|x_k - x^*\|^2 \leq (1 - 1/\kappa)^k R_0^2$.

The optimal GeoD (with the linear convergence rate $(1 - 1/\sqrt{\kappa})$) maintains two balls containing $x^*$ in each iteration, whose centers are $c_k$ and $x_{k+1}^{++}$, respectively. More specifically, suppose that in the $k$-th iteration we have $c_k$ and $x_k$, then $c_{k+1}$ and $x_{k+1}$ are obtained as follows. First, $x_{k+1}$ is the minimizer of $f$ on $\text{Line}(c_k, x_k^+)$. Second, $c_{k+1}$ (resp. $R_{k+1}^2$) is the center (resp. squared radius) of the ball (given by Lemma 2.1) that contains

$$B\big(c_k, R_k^2 - \|\nabla f(x_{k+1})\|^2/(\alpha^2\kappa)\big) \cap B\big(x_{k+1}^{++}, (1 - 1/\kappa)\|\nabla f(x_{k+1})\|^2/\alpha^2\big).$$

Calculating $c_{k+1}$ and $R_{k+1}$ is easy and we refer to Algorithm 1 of [1] for details. By applying Lemma 2.1 with $x_A = c_k$, $r_A = R_k$, $r_B = \|\nabla f(x_{k+1})\|/\alpha$, $\epsilon = 1/\kappa$ and $\delta = \frac{2}{\alpha}(f(x_k^+) - f(x^*))$, we obtain $R_{k+1}^2 = (1 - 1/\sqrt{\kappa})R_k^2$, which further implies $\|x^* - c_k\|^2 \leq (1 - 1/\sqrt{\kappa})^k R_0^2$, *i.e.*, the optimal GeoD converges with the linear rate $(1 - 1/\sqrt{\kappa})$.

**Lemma 2.1** (see [1, 10]). *Fix centers $x_A$, $x_B \in \mathbb{R}^n$ and squared radii $r_A^2, r_B^2 > 0$. Also fix $\epsilon \in (0, 1)$ and suppose $\|x_A - x_B\|^2 \geq r_B^2$. There exists a new center $c \in \mathbb{R}^n$ such that for any $\delta > 0$, we have*

$$B(x_A, r_A^2 - \epsilon r_B^2 - \delta) \cap B\big(x_B, r_B^2(1 - \epsilon) - \delta\big) \subset B\big(c, (1 - \sqrt{\epsilon})r_A^2 - \delta\big).$$

## 3 Geometric Descent Method for Nonsmooth Convex Composite Problems

Drusvyatskiy, Fazel and Roy [10] extended the suboptimal GeoD to solve the composite problem (1.1). However, it was not clear how to extend the optimal GeoD to solve problem (1.1). We resolve this problem in this section.

The following lemma is useful to our analysis. Its proof is in the supplementary material.

**Lemma 3.1.** *Given point $x \in \mathbb{R}^n$ and step size $t \in (0, 1/\beta]$, denote $x^+ = x - tG_t(x)$. The following inequality holds for any $y \in \mathbb{R}^n$:*

$$F(y) \geq F(x^+) + \langle G_t(x), y - x \rangle + \frac{t}{2} \|G_t(x)\|^2 + \frac{\alpha}{2} \|y - x\|^2. \tag{3.1}$$

## 3.1 GeoPG Algorithm

In this subsection, we describe our proposed geometric proximal gradient method (GeoPG) for solving (1.1). Throughout Sections 3.1 and 3.2, $t \in (0, 1/\beta]$ is a fixed scalar. The key observation for designing GeoPG is that in the $k$-th iteration one has to find $x_k$ that lies on $\text{Line}(x_{k-1}^+, c_{k-1})$ such that the following two inequalities hold:

$$F(x_k^+) \leq F(x_{k-1}^+) - \frac{t}{2} \|G_t(x_k)\|^2, \text{ and } \|x_k^{++} - c_{k-1}\|^2 \geq \frac{1}{\alpha^2} \|G_t(x_k)\|^2. \tag{3.2}$$

Intuitively, the first inequality in (3.2) requires that there is a function value reduction on $x_k^+$ from $x_{k-1}^+$, and the second inequality requires that the centers of the two balls are far away from each other so that Lemma 2.1 can be applied.

The following lemma gives a sufficient condition for (3.2). Its proof is in the supplementary material.

**Lemma 3.2.** *(3.2) holds if $x_k$ satisfies*

$$\langle x_k^+ - x_k, x_{k-1}^+ - x_k \rangle \leq 0, \text{ and } \langle x_k^+ - x_k, x_k - c_{k-1} \rangle \geq 0. \tag{3.3}$$

Therefore, we only need to find $x_k$ such that (3.3) holds. To do so, we define the following functions for given $x, c$ ($x \neq c$) and $t \in (0, \beta]$:

$$\phi_{t,x,c}(z) = \langle z^+ - z, x - c \rangle, \forall z \in \mathbb{R}^n, \text{ and } \bar{\phi}_{t,x,c}(s) = \phi_{t,x,c}(x + s(c - x)), \forall s \in \mathbb{R}.$$

The functions $\phi_{t,x,c}(z)$ and $\bar{\phi}_{t,x,c}(s)$ have the following properties. Its proof can be found in the supplementary material.

**Lemma 3.3.** *(i) $\phi_{t,x,c}(z)$ is Lipschitz continuous. (ii) $\bar{\phi}_{t,x,c}(s)$ strictly monotonically increases.*

We are now ready to describe how to find $x_k$ such that (3.3) holds. This is summarized in Lemma 3.4.

**Lemma 3.4.** *The following two ways find $x_k$ satisfying (3.3).*

*(i) If $\bar{\phi}_{t,x_{k-1}^+,c_{k-1}}(1) \leq 0$, then (3.3) holds by setting $x_k := c_{k-1}$; if $\bar{\phi}_{t,x_{k-1}^+,c_{k-1}}(0) \geq 0$, then (3.3) holds by setting $x_k := x_{k-1}^+$; if $\bar{\phi}_{t,x_{k-1}^+,c_{k-1}}(1) > 0$ and $\bar{\phi}_{t,x_{k-1}^+,c_{k-1}}(0) < 0$, then there exists $s \in [0, 1]$ such that $\bar{\phi}_{t,x_{k-1}^+,c_{k-1}}(s) = 0$. As a result, (3.3) holds by setting $x_k := x_{k-1}^+ + s(c_{k-1} - x_{k-1}^+)$.*

*(ii) If $\bar{\phi}_{t,x_{k-1}^+,c_{k-1}}(0) \geq 0$, then (3.3) holds by setting $x_k := x_{k-1}^+$; if $\bar{\phi}_{t,x_{k-1}^+,c_{k-1}}(0) < 0$, then there exists $s \geq 0$ such that $\bar{\phi}_{t,x_{k-1}^+,c_{k-1}}(s) = 0$. As a result, (3.3) holds by setting $x_k := x_{k-1}^+ + s(c_{k-1} - x_{k-1}^+)$.*

*Proof.* Case (i) directly follows from the Mean-Value Theorem. Case (ii) follows from the monotonicity and continuity of $\bar{\phi}_{t,x_{k-1}^+,c_{k-1}}$ from Lemma 3.3. □

It is indeed very easy to find $x_k$ satisfying the two cases in Lemma 3.4, since we are tackling a univariate Lipschitz continuous function $\bar{\phi}_{t,x,c}(s)$. Specifically, for case (i) of Lemma 3.4, we can use the bisection method to find the zero of $\bar{\phi}_{t,x_{k-1}^+,c_{k-1}}$ in the closed interval $[0, 1]$. In practice, we found that the Brent-Dekker method [11, 12] performs much better than the bisection method, so we use the Brent-Dekker method in our numerical experiments. For case (ii) of Lemma 3.4, we can use the semi-smooth Newton method to find the zero of $\bar{\phi}_{t,x_{k-1}^+,c_{k-1}}$ in the interval $[0, +\infty)$.

In our numerical experiments, we implemented the global semi-smooth Newton method [13, 14] and obtained very encouraging results. These two procedures are described in Algorithms 1 and 2, respectively. Based on the discussions above, we know that $x_k$ generated by these two algorithms satisfies (3.3) and hence (3.2).

We are now ready to present our GeoPG algorithm for solving (1.1) as in Algorithm 3.

---
**Algorithm 1** : The first procedure for finding $x_k$ from given $x_{k-1}^+$ and $c_{k-1}$.

---
1: **if** $\langle (x_{k-1}^+)^+ - x_{k-1}^+, x_{k-1}^+ - c_{k-1} \rangle \geq 0$ **then**
2:     set $x_k := x_{k-1}^+$;
3: **else if** $\langle c_{k-1}^+ - c_{k-1}, x_{k-1}^+ - c_{k-1} \rangle \leq 0$ **then**
4:     set $x_k := c_{k-1}$;
5: **else**
6:     use the Brent-Dekker method to find $s \in [0, 1]$ such that $\bar{\phi}_{t, x_{k-1}^+, c_{k-1}}(s) = 0$, and set
        $x_k := x_{k-1}^+ + s(c_{k-1} - x_{k-1}^+)$;
7: **end if**

---

---
**Algorithm 2** : The second procedure for finding $x_k$ from given $x_{k-1}^+$ and $c_{k-1}$.

---
1: **if** $\langle (x_{k-1}^+)^+ - x_{k-1}^+, x_{k-1}^+ - c_{k-1} \rangle \geq 0$ **then**
2:     set $x_k := x_{k-1}^+$;
3: **else**
4:     use the global semi-smooth Newton method [13, 14] to find the root $s \in [0, +\infty)$ of
        $\bar{\phi}_{t, x_{k-1}^+, c_{k-1}}(s)$, and set $x_k := x_{k-1}^+ + s(c_{k-1} - x_{k-1}^+)$;
5: **end if**

---

## 3.2 Convergence Analysis of GeoPG

We are now ready to present our main convergence result for GeoPG.

**Theorem 3.5.** *Given initial point $x_0$ and step size $t \in (0, 1/\beta]$, we set $R_0^2 = \frac{\|G_t(x_0)\|^2}{\alpha^2}(1 - \alpha t)$. Suppose that sequence $\{(x_k, c_k, R_k)\}$ is generated by Algorithm 3, and that $x^*$ is the optimal solution of (1.1) and $F^*$ is the optimal objective value. For any $k \geq 0$, one has $x^* \in B(c_k, R_k^2)$ and $R_{k+1}^2 \leq (1 - \sqrt{\alpha t})R_k^2$, and thus*

$$\|x^* - c_k\|^2 \leq (1 - \sqrt{\alpha t})^k R_0^2, \text{ and } F(x_{k+1}^+) - F^* \leq \frac{\alpha}{2}(1 - \sqrt{\alpha t})^k R_0^2. \tag{3.4}$$

*Note that when $t = 1/\beta$, (3.4) implies the linear convergence rate $(1 - 1/\sqrt{\kappa})$.*

*Proof.* We prove a stronger result by induction that for every $k \geq 0$, one has

$$x^* \in B\big(c_k, R_k^2 - 2(F(x_k^+) - F^*)/\alpha\big). \tag{3.5}$$

Let $y = x^*$ in (3.1). We have $\|x^* - x^{++}\|^2 \leq (1 - \alpha t)\|G_t(x)^2\|/\alpha^2 - 2(F(x^+) - F^*)/\alpha$, implying

$$x^* \in B\big(x^{++}, \|G_t(x)\|^2(1 - \alpha t)/\alpha^2 - 2(F(x^+) - F^*)/\alpha\big). \tag{3.6}$$

Setting $x = x_0$ in (3.6) shows that (3.5) holds for $k = 0$. We now assume that (3.5) holds for some $k \geq 0$, and in the following we will prove that (3.5) holds for $k + 1$. Combining (3.5) and the first inequality of (3.2) yields

$$x^* \in B\big(c_k, R_k^2 - t\|G_t(x_{k+1})\|^2/\alpha - 2(F(x_{k+1}^+) - F^*)/\alpha\big). \tag{3.7}$$

By setting $x = x_{k+1}$ in (3.6), we obtain

$$x^* \in B\big(x_{k+1}^{++}, \|G_t(x_{k+1})\|^2(1 - \alpha t)/\alpha^2 - 2(F(x_{k+1}^+) - F^*)/\alpha\big). \tag{3.8}$$

We now apply Lemma 2.1 to (3.7) and (3.8). Specifically, we set $x_B = x_{k+1}^{++}$, $x_A = c_k$, $\epsilon = \alpha t$, $r_A = R_k$, $r_B = \|G_t(x_{k+1})\|/\alpha$, $\delta = \frac{2}{\alpha}(F(x_k^+) - F^*)$, and note that $\|x_A - x_B\|^2 \geq r_B^2$ because of the second inequality of (3.2). Then Lemma 2.1 indicates that there exists $c_{k+1}$ such that

$$x^* \in B\big(c_{k+1}, (1 - 1/\sqrt{\kappa})R_k^2 - 2(F(x_{k+1}^+) - F^*)/\alpha\big), \tag{3.9}$$

*i.e.*, (3.5) holds for $k + 1$ with $R_{k+1}^2 \leq (1 - \sqrt{\alpha t})R_k^2$. Note that $c_{k+1}$ is the center of the minimum enclosing ball of the intersection of the two balls in (3.7) and (3.8), and can be computed in the same way as Algorithm 1 of [1]. From (3.9) we obtain that $\|x^* - c_{k+1}\|^2 \leq (1 - \sqrt{\alpha t})R_k^2 \leq (1 - \sqrt{\alpha t})^{k+1}R_0^2$. Moreover, (3.7) indicates that $F(x_{k+1}^+) - F^* \leq \frac{\alpha}{2}R_k^2 \leq \frac{\alpha}{2}(1 - \sqrt{\alpha t})^k R_0^2$.   □

---

**Algorithm 3** : GeoPG: geometric proximal gradient descent for convex composite minimization.

---

**Require:** Parameters $\alpha$, $\beta$, initial point $x_0$ and step size $t \in (0, 1/\beta]$.

  1: Set $c_0 = x_0^{++}$, $R_0^2 = \|G_t(x_0)\|^2 (1 - \alpha t)/\alpha^2$;
  2: **for** $k = 1, 2, \ldots$ **do**
  3:     Use Algorithm 1 or 2 to find $x_k$;
  4:     Set $x_A := x_k^{++} = x_k - G_t(x_k)/\alpha$, and $R_A^2 = \|G_t(x_k)\|^2 (1 - \alpha t)/\alpha^2$;
  5:     Set $x_B := c_{k-1}$, and $R_B^2 = R_{k-1}^2 - 2(F(x_{k-1}^+) - F(x_k^+))/\alpha$;
  6:     Compute $B(c_k, R_k^2)$: the minimum enclosing ball of $B(x_A, R_A^2) \cap B(x_B, R_B^2)$, which can be done using Algorithm 1 in [1];
  7: **end for**

---

## 4    Practical Issues

### 4.1    GeoPG with Backtracking

In practice, the Lipschitz constant $\beta$ may be unknown to us. In this subsection, we describe a backtracking strategy for GeoPG in which $\beta$ is not needed. From the $\beta$-smoothness of $f$, we have

$$f(x^+) \leq f(x) - t\langle \nabla f(x), G_t(x) \rangle + t\|G_t(x)\|^2/2. \tag{4.1}$$

Note that inequality (3.1) holds because of (4.1), which holds when $t \in (0, 1/\beta]$. If $\beta$ is unknown, we can perform backtracking on $t$ such that (4.1) holds, which is a common practice for proximal gradient method, *e.g.*, [15–17]. Note that the key step in our analysis of GeoPG is to guarantee that the two inequalities in (3.2) hold. According to Lemma 3.2, the second inequality in (3.2) holds as long as we use Algorithm 1 or Algorithm 2 to find $x_k$, and it does not need the knowledge of $\beta$. However, the first inequality in (3.2) requires $t \leq 1/\beta$, because its proof in Lemma 3.2 needs (3.1). Thus, we need to perform backtracking on $t$ until (4.1) is satisfied, and use the same $t$ to find $x_k$ by Algorithm 1 or Algorithm 2. Our GeoPG algorithm with backtracking (GeoPG-B) is described in Algorithm 4.

---

**Algorithm 4** : GeoPG with Backtracking (GeoPG-B)

---

**Require:** Parameters $\alpha, \gamma \in (0, 1)$, $\eta \in (0, 1)$, initial step size $t_0 > 0$ and initial point $x_0$.

  Repeat $t_0 := \eta t_0$ until (4.1) holds for $t = t_0$;
  Set $c_0 = x_0^{++}$, $R_0^2 = \dfrac{\|G_{t_0}(x_0)\|^2}{\alpha^2}(1 - \alpha t_0)$;
  **for** $k = 1, 2, \ldots$ **do**
    **if** no backtracking was performed in the $(k-1)$-th iteration **then**
        Set $t_k := t_{k-1}/\gamma$;
    **else**
        Set $t_k := t_{k-1}$;
    **end if**
    Compute $x_k$ by Algorithm 1 or Algorithm 2 with $t = t_k$;
    **while** $f(x_k^+) > f(x_k) - t_k\langle \nabla f(x_k), G_{t_k}(x_k) \rangle + \frac{t_k}{2}\|G_{t_k}(x_k)\|^2$ **do**
        Set $t_k := \eta t_k$  (backtracking);
        Compute $x_k$ by Algorithm 1 or Algorithm 2 with $t = t_k$;
    **end while**
    Set $x_A := x_k^{++} = x_k - G_{t_k}(x_k)/\alpha$, $R_A^2 = \dfrac{\|G_{t_k}(x_k)\|^2}{\alpha^2}(1 - \alpha t_k)$;
    Set $x_B := c_{k-1}$, $R_B^2 = R_{k-1}^2 - \frac{2}{\alpha}(F(x_{k-1}^+) - F(x_k^+))$;
    Compute $B(c_k, R_k^2)$: the minimum enclosing ball of $B(x_A, R_A^2) \cap B(x_B, R_B^2)$;
  **end for**

---

Note that the sequence $\{t_k\}$ generated in Algorithm 4 is uniformly bounded away from 0. This is because (4.1) always holds when $t_k \leq 1/\beta$. As a result, we know $t_k \geq t_{\min} := \min_{i=0,\ldots,k} t_i \geq \eta/\beta$. It is easy to see that in the $k$-th iteration of Algorithm 4, $x^*$ is contained in two balls:

$$\begin{aligned}
x^* &\in\; B\big(c_{k-1}, R_{k-1}^2 - t_k\|G_{t_k}(x_k)\|^2/\alpha - 2(F(x_k^+) - F^*)/\alpha\big), \\
x^* &\in\; B\big(x_k^{++}, \|G_{t_k}(x_k)\|^2(1 - \alpha t_k)/\alpha^2 - 2(F(x_k^+) - F^*)/\alpha\big).
\end{aligned}$$

Therefore, we have the following convergence result for Algorithm 4, whose proof is similar to that for Algorithm 3. We thus omit the proof for succinctness.

**Theorem 4.1.** *Suppose that $\{(x_k, c_k, R_k, t_k)\}$ is generated by Algorithm 4. For any $k \geq 0$, one has $x^* \in B(c_k, R_k^2)$ and $R_{k+1}^2 \leq (1 - \sqrt{\alpha t_k})R_k^2$, and thus $\|x^* - c_k\|^2 \leq \prod_{i=0}^{k}(1 - \sqrt{\alpha t_i})^i R_0^2 \leq (1 - \sqrt{\alpha t_{min}})^k R_0^2$.*

### 4.2  GeoPG with Limited Memory

The basic idea of GeoD is that in each iteration we maintain two balls $B(y_1, r_1^2)$ and $B(y_2, r_2^2)$ that both contain $x^*$, and then compute the minimum enclosing ball of their intersection, which is expected to be smaller than both $B(y_1, r_1^2)$ and $B(y_2, r_2^2)$. One very intuitive idea that can possibly improve the performance of GeoD is to maintain more balls from the past, because their intersection should be smaller than the intersection of two balls. This idea has been proposed by [9] and [10]. Specifically, Bubeck and Lee [9] suggested to keep all the balls from past iterations and then compute the minimum enclosing ball of their intersection. For a given bounded set $Q$, the center of its minimum enclosing ball is known as the Chebyshev center, and is defined as the solution to the following problem:

$$\min_{y} \max_{x \in Q} \|y - x\|^2 = \min_{y} \max_{x \in Q} \|y\|^2 - 2y^\top x + \mathrm{Tr}(xx^\top). \tag{4.2}$$

(4.2) is not easy to solve for a general set $Q$. However, when $Q := \cap_{i=1}^{m} B(y_i, r_i^2)$, Beck [18] proved that the relaxed Chebyshev center (RCC) [19], which is a convex quadratic program, is equivalent to (4.2) if $m < n$. Therefore, we can solve (4.2) by solving a convex quadratic program (RCC):

$$\min_{y} \max_{(x,\triangle) \in \Gamma} \|y\|^2 - 2y^\top x + \mathrm{Tr}(\triangle) = \max_{(x,\triangle) \in \Gamma} \min_{y} \|y\|^2 - 2y^\top x + \mathrm{Tr}(\triangle) = \max_{(x,\triangle) \in \Gamma} -\|x\|^2 + \mathrm{Tr}(\triangle), \tag{4.3}$$

where $\Gamma = \{(x, \triangle) : x \in Q, \triangle \succeq xx^\top\}$. If $Q = \cap_{i=1}^{m} B(c_i, r_i^2)$, then the dual of (4.3) is

$$\min \|C\lambda\|^2 - \sum_{i=1}^{m} \lambda_i \|c_i\|^2 + \sum_{i=1}^{m} \lambda_i r_i^2, \text{ s.t. } \sum_{i=1}^{m} \lambda_i = 1, \quad \lambda_i \geq 0, \ i = 1, \ldots, m, \tag{4.4}$$

where $C = [c_1, \ldots, c_m]$ and $\lambda_i, i = 1, 2, \ldots, m$ are the dual variables. Beck [18] proved that the optimal solutions of (4.2) and (4.4) are linked by $x^* = C\lambda^*$ if $m < n$.

Now we can give our limited-memory GeoPG algorithm (L-GeoPG) as in Algorithm 5.

---

**Algorithm 5** : L-GeoPG: Limited-memory GeoPG

---

**Require:** Parameters $\alpha, \beta$, memory size $m > 0$ and initial point $x_0$.
1:  Set $c_0 = x_0^{++}$, $r_0^2 = R_0^2 = \|G_t(x_0)\|^2 (1 - 1/\kappa)/\alpha^2$, and $t = 1/\beta$;
2:  **for** $k = 1, 2, \ldots$ **do**
3:    Use Algorithm 1 or 2 to find $x_k$;
4:    Compute $r_k^2 = \|G_t(x_k)\|^2 (1 - 1/\kappa)/\alpha^2$;
5:    Compute $B(c_k, R_k^2)$: an enclosing ball of the intersection of $B(c_{k-1}, R_{k-1}^2)$ and $Q_k := \cap_{i=k-m+1}^{k} B(x_i^{++}, r_i^2)$ (if $k \leq m$, then set $Q_k := \cap_{i=1}^{k} B(x_i^{++}, r_i^2)$). This is done by setting $c_k = C\lambda^*$, where $\lambda^*$ is the optimal solution of (4.4);
6:  **end for**

---

**Remark 4.2.** *Backtracking can also be incorporated into L-GeoPG. We denote the resulting algorithm as L-GeoPG-B.*

L-GeoPG has the same linear convergence rate as GeoPG, as we show in Theorem 4.3.

**Theorem 4.3.** *Consider the L-GeoPG algorithm. For any $k \geq 0$, one has $x^* \in B(c_k, R_k^2)$ and $R_k^2 \leq (1 - 1/\sqrt{\kappa})R_{k-1}^2$, and thus $\|x^* - c_k\|^2 \leq (1 - 1/\sqrt{\kappa})^k R_0^2$.*

*Proof.* Note that $Q_k := \cap_{i=k-m+1}^{k} B(x_i^{++}, r_i^2) \subset B(x_k^{++}, r_k^2)$. Thus, the minimum enclosing ball of $B(c_{k-1}, R_{k-1}^2) \cap B(x_k^{++}, r_k^2)$ is an enclosing ball of $B(c_{k-1}, R_{k-1}^2) \cap Q_k$. The proof then follows from the proof of Theorem 3.5, and we omit it for brevity. $\square$

# 5 Numerical Experiments

In this section, we compare our GeoPG algorithm with Nesterov's accelerated proximal gradient (APG) method for solving two nonsmooth problems: linear regression and logistic regression, both with elastic net regularization. Because of the elastic net term, the strong convexity parameter $\alpha$ is known. However, we assume that $\beta$ is unknown, and implement backtracking for both GeoPG and APG, *i.e.*, we test GeoPG-B and APG-B (APG with backtracking). We do not target at comparing with other efficient algorithms for solving these two problems. Our main purpose here is to illustrate the performance of this new first-order method GeoPG. Further improvement of this algorithm and comparison with other state-of-the-art methods will be a future research topic.

The initial points were set to zero. To obtain the optimal objective function value $F^*$, we ran APG-B and GeoPG-B for a sufficiently long time and the smaller function value returned by the two algorithms is selected as $F^*$. APG-B was terminated if $(F(x_k) - F^*)/F^* \leq tol$, and GeoPG-B was terminated if $(F(x_k^+) - F^*)/F^* \leq tol$, where $tol = 10^{-8}$ is the accuracy tolerance. The parameters used in backtracking were set to $\eta = 0.5$ and $\gamma = 0.9$. In GeoPG-B, we used Algorithm 2 to find $x_k$, because we found that the performance of Algorithm 2 is slightly better than Algorithm 1 in practice. In the experiments, we ran Algorithm 2 until the absolute value of $\bar{\phi}$ is smaller than $10^{-8}$. The code was written in Matlab and run on a standard PC with 3.20 GHz I5 Intel microprocessor and 16GB of memory. In all figures we reported, the $x$-axis denotes the CPU time (in seconds) and $y$-axis denotes $(F(x_k^+) - F^*)/F^*$.

## 5.1 Linear regression with elastic net regularization

In this subsection, we compare GeoPG-B and APG-B in terms of solving linear regression with elastic net regularization, a popular problem in machine learning and statistics [20]:

$$\min_{x\in\mathbb{R}^n} \ \frac{1}{2p}\|Ax - b\|^2 + \frac{\alpha}{2}\|x\|^2 + \mu\|x\|_1, \tag{5.1}$$

where $A \in \mathbb{R}^{p\times n}$, $b \in \mathbb{R}^p$, and $\alpha, \mu > 0$ are the weighting parameters.

We conducted tests on two real datasets downloaded from the LIBSVM repository: a9a, RCV1. The results are reported in Figure 1. In particular, we tested $\alpha = 10^{-8}$ and $\mu = 10^{-3}, 10^{-4}, 10^{-5}$. Note that since $\alpha$ is very small, the problems are very likely to be ill-conditioned. We see from Figure 1 that GeoPG-B is faster than APG-B on these real datasets, which indicates that GeoPG-B is preferable than APG-B. In the supplementary material, we show more numerical results on varying $\alpha$, which further confirm that GeoPG-B is faster than APG-B when the problems are more ill-conditioned.

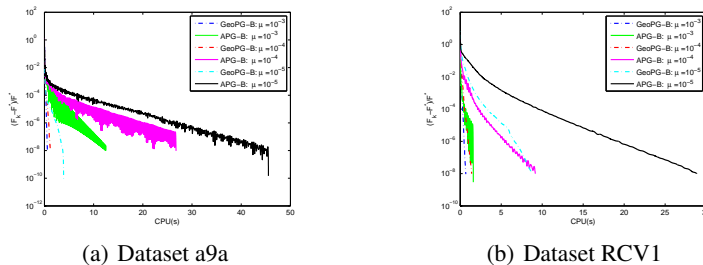

(a) Dataset a9a          (b) Dataset RCV1

Figure 1: GeoPG-B and APG-B for solving (5.1) with $\alpha = 10^{-8}$.

## 5.2 Logistic regression with elastic net regularization

In this subsection, we compare the performance of GeoPG-B and APG-B in terms of solving the following logistic regression problem with elastic net regularization:

$$\min_{x\in\mathbb{R}^n} \ \frac{1}{p}\sum_{i=1}^{p}\log\left(1 + \exp(-b_i \cdot a_i^\top x)\right) + \frac{\alpha}{2}\|x\|^2 + \mu\|x\|_1, \tag{5.2}$$

where $a_i \in \mathbb{R}^n$ and $b_i \in \{\pm 1\}$ are the feature vector and class label of the $i$-th sample, respectively, and $\alpha, \mu > 0$ are the weighting parameters.

We tested GeoPG-B and APG-B for solving (5.2) on the three real datasets a9a, RCV1 and Gisette from LIBSVM, and the results are reported in Figure 2. In particular, we tested $\alpha = 10^{-8}$ and $\mu = 10^{-3}, 10^{-4}, 10^{-5}$. Figure 2 shows that with the same $\mu$, GeoPG-B is much faster than APG-B. More numerical results are provided in the supplementary material, which also indicate that GeoPG-B is much faster than APG-B, especially when the problems are more ill-conditioned.

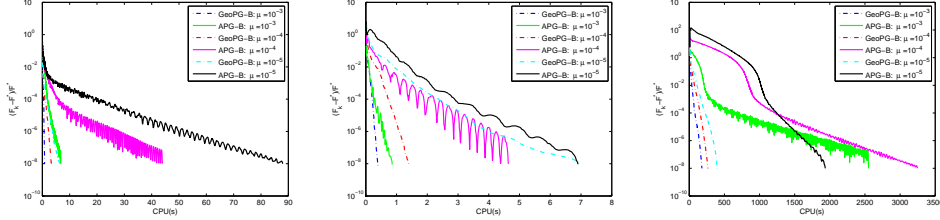

Figure 2: GeoPG-B and APG-B for solving (5.2) with $\alpha = 10^{-8}$. Left: dataset a9a; Middle: dataset RCV1; Right: dataset Gisette.

## 5.3 Numerical results of L-GeoPG-B

In this subsection, we test GeoPG with limited memory described in Algorithm 5 in solving (5.2) on the Gisette dataset. Since we still need to use the backtracking technique, we actually tested L-GeoPG-B. The results with different memory sizes $m$ are reported in Figure 3. Note that $m = 0$ corresponds to the original GeoPG-B without memory. The subproblem (4.4) is solved using the function "quadprog" in Matlab. From Figure 3 we see that roughly speaking, L-GeoPG-B performs better for larger memory sizes, and in most cases, the performance of L-GeoPG-B with $m = 100$ is the best among the reported results. This indicates that the limited-memory idea indeed helps improve the performance of GeoPG.

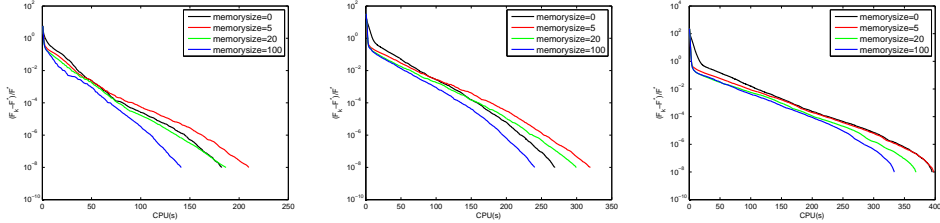

Figure 3: L-GeoPG-B for solving (5.2) on the dataset Gisette with $\alpha = 10^{-8}$. Left: $\mu = 10^{-3}$; Middle: $\mu = 10^{-4}$; Right: $\mu = 10^{-5}$.

## 6 Conclusions

In this paper, we proposed a GeoPG algorithm for solving nonsmooth convex composite problems, which is an extension of the recent method GeoD that can only handle smooth problems. We proved that GeoPG enjoys the same optimal rate as Nesterov's accelerated gradient method for solving strongly convex problems. The backtracking technique was adopted to deal with the case when the Lipschitz constant is unknown. Limited-memory GeoPG was also developed to improve the practical performance of GeoPG. Numerical results on linear regression and logistic regression with elastic net regularization demonstrated the efficiency of GeoPG. It would be interesting to see how to extend GeoD and GeoPG to tackle non-strongly convex problems, and how to further accelerate the running time of GeoPG. We leave these questions in future work.

**Acknowledgements.** Shixiang Chen is supported by CUHK Research Postgraduate Student Grant for Overseas Academic Activities. Shiqian Ma is supported by a startup funding in UC Davis.

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
