[Supplementary Material]

# Supplementary Material: Geometric Descent Method for Convex Composite Minimization

**Shixiang Chen**[1]**, Shiqian Ma**[2]**, and Wei Liu**[3]

[1]Department of SEEM, The Chinese University of Hong Kong, Hong Kong
[2]Department of Mathematics, UC Davis, USA
[3]Tencent AI Lab, China

## 1   Geometric Interpretation of GeoPG

We argue that the geometric intuition of GeoPG is still clear. Note that we are still constructing two balls that contain $x^*$ and shrink at the same absolute amount. In GeoPG, since we assume that the smooth function $f$ is strongly convex, we naturally have one ball that contains $x^*$, and this ball is related to the proximal gradient $G_t$, instead of the gradient due to the presence of the nonsmooth function $h$. To construct the other ball, GeoD needs to perform an exact line search, while our GeoPG needs to find the root of a newly constructed function $\bar{\phi}$, which is again due to the presence of the nonsmooth function $h$. The two changes of GeoPG from GeoD are: replace gradient by proximal gradient; replace the exact line search by finding the root of $\bar{\phi}$, both of which are resulted by the presence of the nonsmooth function $h$.

## 2   Proofs

### 2.1   Proof of Lemma 3.1

*Proof.*   From the $\beta$-smoothness of $f$, we have

$$f(x^+) \leq f(x) - t\langle \nabla f(x), G_t(x)\rangle + \frac{t}{2}\|G_t(x)\|^2. \tag{2.1}$$

Combining (2.1) with

$$f(x) + \langle \nabla f(x), y - x\rangle + \frac{\alpha}{2}\|y - x\|^2 \leq f(y), \ \forall x, y \in \mathbb{R}^n, \tag{2.2}$$

yields that

$$\begin{aligned}
F(x^+) &\leq f(y) - \langle \nabla f(x), y - x\rangle - \frac{\alpha}{2}\|y - x\|^2 - t\langle \nabla f(x), G_t(x)\rangle + \frac{t}{2}\|G_t(x)\|^2 + h(x^+) \\
&= F(y) - \frac{\alpha}{2}\|y - x\|^2 + \frac{t}{2}\|G_t(x)\|^2 + h(x^+) - h(y) - \langle \nabla f(x) - G_t(x), y - x^+\rangle - \langle G_t(x), y - x^+\rangle \\
&\leq F(y) - \frac{\alpha}{2}\|y - x\|^2 + \frac{t}{2}\|G_t(x)\|^2 - \langle G_t(x), y - x^+\rangle,
\end{aligned}$$

$$\tag{2.3}$$

where the last inequality is due to the convexity of $h$ and $G_t(x) \in \nabla f(x) + \partial h(x^+)$.   □

### 2.2   Proof of Lemma 3.2

*Proof.*   Assume

$$\langle x_k^+ - x_k, x_{k-1}^+ - x_k\rangle \leq 0, \text{ and } \langle x_k^+ - x_k, x_k - c_{k-1}\rangle \geq 0, \tag{2.4}$$

holds. By letting $y = x_{k-1}^+$ and $x = x_k$ in (2.3), we have

$$
\begin{aligned}
F(x_k^+) &\le F(x_{k-1}^+) - \langle G_t(x_k), x_{k-1}^+ - x_k \rangle - \frac{t}{2}\|G_t(x_k)\|^2 - \frac{\alpha}{2}\|x_{k-1}^+ - x_k\|^2 \\
&= F(x_{k-1}^+) + \frac{1}{t}\langle x_k^+ - x_k, x_{k-1}^+ - x_k \rangle - \frac{t}{2}\|G_t(x_k)\|^2 - \frac{\alpha}{2}\|x_{k-1}^+ - x_k\|^2 \\
&\le F(x_{k-1}^+) - \frac{t}{2}\|G_t(x_k)\|^2,
\end{aligned}
$$

where the last inequality is due to (2.4). Moreover, from the definition of $x_k^{++}$ and (2.4) it is easy to see

$$
\|x_k^{++} - c_{k-1}\|^2 = \|x_k - c_{k-1}\|^2 + \frac{2}{\alpha t}\langle x_k^+ - x_k, x_k - c_{k-1}\rangle + \frac{1}{\alpha^2}\|G_t(x_k)\|^2 \ge \frac{1}{\alpha^2}\|G_t(x_k)\|^2.
$$

$\square$

## 2.3 Proof of Lemma 3.3

Before we prove Lemma 3.3, we need the following well-know result, which can be found in [2].

**Lemma.** (see Lemma 3.9 of [2]) For $t \in (0, 1/\beta]$, $G_t(x)$ is strongly monotone, i.e.,

$$
\langle G_t(x) - G_t(y), x - y \rangle \ge \frac{\alpha}{2}\|x - y\|^2, \forall x, y. \tag{2.5}
$$

We are now ready to prove Lemma 3.3.

*Proof.* We prove (i) first.

$$
\begin{aligned}
|\phi_{t,x,c}(z_1) - \phi_{t,x,c}(z_2)| &= |\langle z_1^+ - z_1 - (z_2^+ - z_2), x - c\rangle| \le \|z_1^+ - z_2^+ - (z_1 - z_2)\|\|x - c\| \\
&\le (\|\text{prox}_{th}(z_1 - t\nabla f(z_1)) - \text{prox}_{th}(z_2 - t\nabla f(z_2))\| + \|z_1 - z_2\|)\|x - c\| \\
&\le (2 + t\beta)\|x - c\|\|z_1 - z_2\|,
\end{aligned}
$$

where the last inequality is due to the non-expansiveness of the proximal mapping operation.

We now prove (ii). For $s_1 < s_2$, let $z_1 = x + s_1(c - x)$ and $z_2 = x + s_2(c - x)$. We have

$$
\begin{aligned}
\bar{\phi}_{t,x,c}(s_2) - \bar{\phi}_{t,x,c}(s_1) &= \langle z_2^+ - z_2 - (z_1^+ - z_1), x - c\rangle = \frac{t}{s_2 - s_1}\langle G_t(z_2) - G_t(z_1), z_2 - z_1\rangle \\
&\ge \frac{\alpha t}{2}(s_2 - s_1)\|x - c\|^2 > 0,
\end{aligned}
$$

where the first inequality follows from (2.5). $\square$

# 3 Numerical Experiment on Other Datasets

In this section, we report some numerical results of other data sets. Here we set the terminate condition as $\|G_t(x_k^+)\|_\infty \le tol$ for GeoP-B and $\|G_t(x_k)\|_\infty \le tol$ for APG-B.

## 3.1 Linear regression with elastic net regularization

In this subsection, we compare GeoPG-B and APG-B for solving linear regression with elastic net regularization:

$$
\min_{x \in \mathbb{R}^n} \frac{1}{2p}\|Ax - b\|^2 + \frac{\alpha}{2}\|x\|^2 + \mu\|x\|_1, \tag{3.1}
$$

where $A \in \mathbb{R}^{p \times n}$, $b \in \mathbb{R}^p$, $\alpha, \mu > 0$ are weighting parameters.

We first compare these two algorithms on some synthetic data. In our experiments, entries of $A$ were drawn randomly from the standard Gaussian distribution, the solution $\bar{x}$ was a sparse vector with 10% nonzero entries whose locations are uniformly random and whose values follow the Gaussian distribution $3 * \mathcal{N}(0, 1)$, and $b = A * \bar{x} + \mathbf{n}$, where the noise $\mathbf{n}$ follows the Gaussian distribution

$0.01 * \mathcal{N}(0,1)$. Moreover, since we assume that the strong convexity parameter of (3.1) is equal to $\alpha$, when $p > n$, we manipulate $A$ such that the smallest eigenvalue of $A^\top A$ is equal to 0. Specifically, when $p > n$, we truncate the smallest eigenvalue of $A^\top A$ to 0, and obtain the new $A$ by eigenvalue decomposition of $A^\top A$. We set $tol = 10^{-8}$.

In Tables 1, 2 and 3, we report the comparison results of GeoPG-B and APG-B for solving different instances of (3.1). We use "f-ev", "g-ev", "p-ev" and "MVM" to denote the number of evaluations of objective function, gradient, proximal mapping of $\ell_1$ norm, and matrix-vector multiplications, respectively. The CPU times are in seconds. We use "–" to denote that the algorithm does not converge in $10^5$ iterations. We tested different values of $\alpha$, which reflect different condition numbers of the problem. We also tested different values of $\mu$, which was set to $\mu = (10^{-3}, 10^{-4}, 10^{-5})/p \times \|A^\top b\|_\infty$, respectively. "f-diff" denotes the absolute difference of the objective values returned by the two algorithms.

From Tables 1, 2 and 3 we see that GeoPG-B is more efficient than APG-B in terms of CPU time when $\alpha$ is small. For example, Table 1 indicates that GeoPG-B is faster than APG-B when $\alpha \le 10^{-4}$, Table 2 indicates that GeoPG-B is faster than APG-B when $\alpha \le 10^{-6}$, and Table 3 shows that GeoPG-B is faster than APG-B when $\alpha \le 10^{-8}$. Since a small $\alpha$ corresponds to a large condition number, we can conclude that in this case GeoPG-B is more preferable than APG-B for ill-conditioned problems. Note that "f-diff" is very small in all cases, which indicates that the solutions returned by GeoPG-B and APG-B are very close.

We also conducted tests on three real datasets downloaded from the LIBSVM repository: a9a, RCV1 and Gisette, among which a9a and RCV1 are sparse and Gisette is dense. The size and sparsity (percentage of nonzero entries) of these three datasets are $(32561 \times 123, 11.28\%)$, $(20242 \times 47236, 0.16\%)$ and $(6000 \times 5000, 99.1\%)$, respectively. The results are reported in Tables 4, 5 and 6, where $\alpha = 10^{-2}, 10^{-4}, 10^{-6}, 10^{-8}, 10^{-10}$ and $\mu = 10^{-3}, 10^{-4}, 10^{-5}$. We see from these tables that GeoPG-B is faster than APG-B on these real datasets when $\alpha$ is small, i.e., when the problem is more ill-conditioned.

### 3.2 Logistic regression with elastic net regularization

In this subsection, we compare the performance of GeoPG-B and APG-B for solving the following logistic regression problem with elastic net regularization:

$$\min_{x \in \mathbb{R}^n} \frac{1}{p} \sum_{i=1}^{p} \log(1 + \exp(-b_i \cdot a_i^\top x)) + \frac{\alpha}{2}\|x\|^2 + \mu\|x\|_1, \qquad (3.2)$$

where $a_i \in \mathbb{R}^n$ and $b_i \in \{\pm 1\}$ are the feature vector and class label of the $i$-th sample, respectively, and $\alpha, \mu > 0$ are weighting parameters.

We first compare GeoPG-B and APG-B for solving (3.2) on some synthetic data. In our experiments, each $a_i$ was drawn randomly from the standard Gaussian distribution, the linear model parameter $\bar{x}$ was a sparse vector with 10% nonzero entries whose locations are uniformly random and whose values follow the Gaussian distribution $3 * \mathcal{N}(0,1)$, and $\ell = A * \bar{x} + \mathbf{n}$, where noise $\mathbf{n}$ follows the Gaussian distribution $0.01 * \mathcal{N}(0,1)$. Then, we generate class labels as bernoulli random variables with the parameter $1/(1 + \exp \ell_i)$. We set $tol = 10^{-8}$.

In Tables 7, 8 and 9 we report the comparison results of GeoPG-B and APG-B for solving different instances of (3.2). From results in these tables we again observe that GeoPG-B is faster than APG-B when $\alpha$ is small, i.e., when the condition number is large.

We also tested GeoPG-B and APG-B for solving (3.2) on the three real datasets a9a, RCV1 and Gisette from LIBSVM, and the results are reported in Tables 10, 11 and 12. We again have the similar observations as before, i.e., GeoPG-B is faster than APG-B for more ill-conditioned problems.

### 3.3 More discussions on the numerical results

To the best of our knowledge, the FISTA algorithm [1] does not have a counterpart for strongly convex problem, but we still conducted some numerical experiments using FISTA for solving the above problems. We found that FISTA and APG are comparable, but they are both worse than GeoPG for more ill-conditioned problems. Moreover, from the results in this section, we can see that when

the problem is well-posed such as $\alpha = 0.01$, APG is usually faster than GeoPG in the CPU time, and when the problem is ill-posed such as $\alpha = 10^{-6}, 10^{-8}, 10^{-10}$, GeoPG is usually faster, but the iterate of GeoPG is less than APG in the most cases. So GeoPG is not always better than APG in the CPU time. But since ill-posed problems are more challenging to solve, we believe that these numerical results showed the potential of GeoPG. The reason why GeoPG is better than APG for ill-posed problem is still not clear at this moment, but we think that it might be related to the fact that APG is not monotone but GeoPG is, which can be seen from the figures in our paper. Furthermore, although GeoPG requires to find the root of a function $\bar{\phi}$ in each iteration, we found that a very good approximation of the root can be obtained by running the semi-smooth Newton method for 1-2 iterations on average. This explains why these steps of GeoPG do not bring much trouble in practice.

### 3.4 Numerical results of L-GeoPG-B

In this subsection, we tested GeoPG-B with limited memory described in Algorithm 5 on solving (3.2) on Gisette dataset. The results for different memory size $m$ are reported in Table 13. Note that $m = 0$ corresponds to the original GeoPG-B without memory.

From Table 13 we see that roughly speaking, L-GeoPG-B performs better for larger memory size, and in almost all cases, the performance of L-GeoPG-B with $m = 100$ is the best among the reported results. This indicates that the limited-memory idea indeed helps improve the performance of GeoPG.

Table 1: GeoPG-B and APG-B for solving linear regression with elastic net regularization. $p = 4000, n = 2000$

| | APG-B | | | | | | GeoPG-B | | | | | | |
|---|---|---|---|---|---|---|---|---|---|---|---|---|---|
| $\alpha$ | iter | cpu | f-ev | g-ev | p-ev | MVM | iter | cpu | f-ev | g-ev | p-ev | MVM | f-diff |
| $\mu = 1.136e - 02$ | | | | | | | | | | | | | |
| $10^{-2}$ | 172 | 1.0 | 354 | 326 | 194 | 384 | 156 | 1.1 | 457 | 348 | 352 | 398 | 8.5e-14 |
| $10^{-4}$ | 538 | 2.8 | 1116 | 1020 | 611 | 1203 | 95 | 0.7 | 267 | 240 | 245 | 247 | 6.4e-14 |
| $10^{-6}$ | 905 | 4.9 | 1868 | 1715 | 1029 | 2030 | 94 | 0.7 | 260 | 249 | 254 | 247 | 5.0e-14 |
| $10^{-8}$ | 1040 | 5.4 | 2146 | 2003 | 1182 | 2332 | 95 | 0.7 | 263 | 258 | 263 | 247 | 1.4e-14 |
| $10^{-10}$ | 964 | 5.0 | 2002 | 1805 | 1095 | 2154 | 95 | 0.7 | 263 | 267 | 272 | 247 | 2.1e-14 |
| $\mu = 1.136e - 03$ | | | | | | | | | | | | | |
| $10^{-2}$ | 175 | 0.9 | 356 | 332 | 197 | 392 | 168 | 1.2 | 493 | 384 | 388 | 432 | 1.3e-13 |
| $10^{-4}$ | 687 | 3.6 | 1414 | 1304 | 779 | 1539 | 145 | 1.0 | 411 | 392 | 397 | 377 | 1.5e-14 |
| $10^{-6}$ | 999 | 5.1 | 2086 | 1676 | 1134 | 2225 | 140 | 1.0 | 371 | 384 | 394 | 354 | 6.5e-14 |
| $10^{-8}$ | 1122 | 5.8 | 2348 | 1827 | 1275 | 2499 | 143 | 1.0 | 374 | 420 | 429 | 365 | 1.8e-15 |
| $10^{-10}$ | 1142 | 5.9 | 2388 | 1858 | 1298 | 2545 | 143 | 1.0 | 374 | 449 | 458 | 365 | 6.2e-15 |
| $\mu = 1.136e - 04$ | | | | | | | | | | | | | |
| $10^{-2}$ | 168 | 0.9 | 346 | 314 | 189 | 374 | 113 | 0.8 | 328 | 252 | 256 | 296 | 1.4e-14 |
| $10^{-4}$ | 911 | 4.8 | 1836 | 1853 | 1035 | 2064 | 207 | 1.5 | 603 | 587 | 592 | 535 | 4.1e-14 |
| $10^{-6}$ | 2293 | 11.9 | 4744 | 3936 | 2605 | 5132 | 191 | 1.4 | 523 | 596 | 602 | 492 | 3.8e-14 |
| $10^{-8}$ | 3979 | 20.5 | 8266 | 5923 | 4526 | 8899 | 199 | 1.4 | 500 | 713 | 728 | 501 | 9.8e-14 |
| $10^{-10}$ | 4503 | 23.3 | 9364 | 6668 | 5123 | 10068 | 185 | 1.3 | 456 | 624 | 639 | 465 | 5.9e-14 |

Table 2: GeoPG-B and APG-B for solving linear regression with elastic net regularization. $p = 2000, n = 2000$

| | APG-B | | | | | | GeoPG-B | | | | | | |
|---|---|---|---|---|---|---|---|---|---|---|---|---|---|
| $\alpha$ | iter | cpu | f-ev | g-ev | p-ev | MVM | iter | cpu | f-ev | g-ev | p-ev | MVM | f-diff |
| $\mu = 1.50e - 02$ | | | | | | | | | | | | | |
| $10^{-2}$ | 244 | 0.7 | 498 | 475 | 276 | 548 | 304 | 1.3 | 889 | 690 | 694 | 774 | 3.4e-13 |
| $10^{-4}$ | 1800 | 4.8 | 3690 | 3582 | 2046 | 4048 | 545 | 2.4 | 1569 | 1298 | 1308 | 1378 | 1.3e-12 |
| $10^{-6}$ | 9706 | 26.0 | 19722 | 20445 | 11040 | 21926 | 557 | 2.3 | 1598 | 1328 | 1339 | 1415 | 2.8e-12 |
| $10^{-8}$ | 20056 | 53.7 | 40528 | 43361 | 22817 | 45427 | 561 | 2.3 | 1614 | 1332 | 1344 | 1416 | 2.4e-12 |
| $10^{-10}$ | 20473 | 53.9 | 41426 | 44159 | 23298 | 46357 | 565 | 2.3 | 1626 | 1373 | 1385 | 1436 | 2.4e-12 |
| $\mu = 1.50e - 03$ | | | | | | | | | | | | | |
| $10^{-2}$ | 241 | 0.6 | 496 | 463 | 273 | 540 | 280 | 1.2 | 813 | 634 | 638 | 716 | 1.4e-14 |
| $10^{-4}$ | 1926 | 5.1 | 3968 | 3708 | 2188 | 4319 | 1218 | 5.0 | 3560 | 2875 | 2892 | 3073 | 2.0e-11 |
| $10^{-6}$ | 12502 | 32.7 | 25658 | 24681 | 14222 | 28118 | 1297 | 5.3 | 3718 | 3065 | 3097 | 3262 | 1.1e-11 |
| $10^{-8}$ | 47139 | 124.3 | 95560 | 100584 | 53646 | 106652 | 1289 | 5.3 | 3686 | 3043 | 3074 | 3245 | 2.1e-11 |
| $10^{-10}$ | 72186 | 194.3 | 145934 | 156713 | 82157 | 163534 | 1297 | 5.2 | 3717 | 3098 | 3132 | 3262 | 2.5e-11 |
| $\mu = 1.50e - 04$ | | | | | | | | | | | | | |
| $10^{-2}$ | 239 | 0.6 | 488 | 460 | 270 | 536 | 225 | 0.9 | 648 | 510 | 514 | 584 | 3.3e-13 |
| $10^{-4}$ | 1985 | 5.2 | 4048 | 3860 | 2257 | 4476 | 1713 | 6.9 | 5041 | 4040 | 4058 | 4322 | 7.0e-11 |
| $10^{-6}$ | 13824 | 35.7 | 28534 | 25354 | 15726 | 31010 | 2527 | 10.2 | 7225 | 6019 | 6082 | 6345 | 2.5e-11 |
| $10^{-8}$ | 56339 | 146.2 | 116280 | 106460 | 64105 | 126410 | 2594 | 10.6 | 7288 | 6095 | 6182 | 6491 | 3.6e-11 |
| $10^{-10}$ | – | – | – | – | – | – | 2573 | 10.4 | 7217 | 6075 | 6163 | 6446 | – |

Table 3: GeoPG-B and APG-B for solving linear regression with elastic net regularization. $p = 2000, n = 4000$

| | APG-B | | | | | | GeoPG-B | | | | | | |
|---|---|---|---|---|---|---|---|---|---|---|---|---|---|
| $\alpha$ | iter | cpu | f-ev | g-ev | p-ev | MVM | iter | cpu | f-ev | g-ev | p-ev | MVM | f-diff |
| $\mu = 1.82e - 02$ | | | | | | | | | | | | | |
| $10^{-2}$ | 327 | 1.9 | 660 | 680 | 371 | 740 | 387 | 2.8 | 1117 | 936 | 946 | 980 | 2.0e-13 |
| $10^{-4}$ | 2263 | 12.8 | 4620 | 4445 | 2571 | 5096 | 2454 | 17.9 | 6858 | 6181 | 6225 | 6168 | 4.3e-11 |
| $10^{-6}$ | 12579 | 67.5 | 25566 | 26229 | 14312 | 28421 | 4478 | 32.7 | 12494 | 11180 | 11216 | 11300 | 1.8e-11 |
| $10^{-8}$ | 55577 | 299.3 | 112140 | 121939 | 63268 | 126044 | 4595 | 33.7 | 12814 | 11754 | 11795 | 11609 | 1.4e-10 |
| $10^{-10}$ | – | – | – | – | – | – | 4645 | 34.6 | 13204 | 12088 | 12129 | 11729 | – |
| $\mu = 1.82e - 03$ | | | | | | | | | | | | | |
| $10^{-2}$ | 306 | 1.7 | 622 | 621 | 346 | 688 | 279 | 2.1 | 813 | 677 | 684 | 713 | 6.4e-13 |
| $10^{-4}$ | 2355 | 12.7 | 4820 | 4534 | 2675 | 5296 | 2634 | 19.3 | 7482 | 6774 | 6846 | 6596 | 3.9e-13 |
| $10^{-6}$ | 14827 | 79.8 | 30328 | 28671 | 16862 | 33388 | 12756 | 94.1 | 36510 | 32580 | 32735 | 32121 | 2.2e-10 |
| $10^{-8}$ | 56286 | 305.7 | 114576 | 115199 | 64050 | 127099 | 11665 | 88.0 | 32397 | 32580 | 31987 | 29352 | 6.1e-11 |
| $10^{-10}$ | – | – | – | – | – | – | 13830 | 102.4 | 38547 | 37931 | 38088 | 34885 | – |
| $\mu = 1.82e - 04$ | | | | | | | | | | | | | |
| $10^{-2}$ | 283 | 1.5 | 576 | 560 | 320 | 636 | 219 | 1.6 | 643 | 523 | 528 | 561 | 4.7e-13 |
| $10^{-4}$ | 2420 | 13.2 | 4864 | 5242 | 2749 | 5487 | 2339 | 17.2 | 6818 | 6467 | 6509 | 5882 | 5.8e-11 |
| $10^{-6}$ | 16882 | 91.4 | 34412 | 31337 | 19186 | 38049 | 14803 | 109.3 | 41943 | 44052 | 44384 | 37152 | 4.9e-10 |
| $10^{-8}$ | 79693 | 430.5 | 163098 | 146951 | 90639 | 179423 | 41331 | 305.8 | 116983 | 113344 | 113952 | 104206 | 1.6e-10 |
| $10^{-10}$ | – | – | – | – | – | – | 47501 | 350.2 | 129513 | 151332 | 152224 | 119660 | – |

Table 4: GeoPG-B and APG-B for solving linear regression with elastic net regularization on dataset a9a

| | APG-B | | | | | | GeoPG-B | | | | | | |
|---|---|---|---|---|---|---|---|---|---|---|---|---|---|
| $\alpha$ | iter | cpu | f-ev | g-ev | p-ev | MVM | iter | cpu | f-ev | g-ev | p-ev | MVM | f-diff |
| $\lambda = 1e-03$ | | | | | | | | | | | | | |
| $10^{-2}$ | 266 | 0.3 | 540 | 530 | 301 | 599 | 260 | 0.6 | 769 | 602 | 608 | 662 | 1.3e-14 |
| $10^{-4}$ | 1758 | 1.7 | 3562 | 3683 | 1998 | 3974 | 463 | 1.1 | 1374 | 1138 | 1144 | 1196 | 1.2e-14 |
| $10^{-6}$ | 10790 | 10.4 | 21654 | 23858 | 12277 | 24518 | 410 | 0.9 | 1216 | 964 | 970 | 1058 | 1.5e-13 |
| $10^{-8}$ | 23279 | 22.2 | 46646 | 52163 | 26493 | 52943 | 412 | 0.9 | 1222 | 976 | 982 | 1060 | 1.9e-13 |
| $10^{-10}$ | 26057 | 24.9 | 52236 | 58464 | 29660 | 59260 | 431 | 0.9 | 1279 | 1063 | 1069 | 1104 | 2.2e-13 |
| $\lambda = 1e-04$ | | | | | | | | | | | | | |
| $10^{-2}$ | 267 | 0.3 | 544 | 526 | 302 | 600 | 249 | 0.5 | 734 | 571 | 577 | 642 | 6.7e-16 |
| $10^{-4}$ | 1948 | 1.9 | 3934 | 4100 | 2214 | 4410 | 1587 | 3.4 | 4747 | 3946 | 3951 | 4025 | 2.9e-12 |
| $10^{-6}$ | 14954 | 14.3 | 30012 | 33215 | 17018 | 33985 | 4801 | 10.4 | 14388 | 11381 | 11386 | 12223 | 1.4e-12 |
| $10^{-8}$ | 63920 | 60.9 | 127954 | 144494 | 72741 | 145426 | 910 | 2.0 | 2715 | 2629 | 2634 | 2347 | 3.7e-12 |
| $10^{-10}$ | 94861 | 90.6 | 189814 | 214931 | 107970 | 215895 | 910 | 2.0 | 2715 | 2441 | 2446 | 2333 | 7.0e-13 |
| $\lambda = 1e-05$ | | | | | | | | | | | | | |
| $10^{-2}$ | 258 | 0.3 | 518 | 507 | 292 | 584 | 235 | 0.5 | 692 | 596 | 602 | 604 | 1.2e-14 |
| $10^{-4}$ | 2035 | 1.9 | 4088 | 4319 | 2315 | 4622 | 1701 | 3.7 | 5090 | 4267 | 4273 | 4312 | 3.7e-12 |
| $10^{-6}$ | 16353 | 15.6 | 32768 | 36396 | 18609 | 37188 | 5773 | 12.5 | 17306 | 14961 | 14967 | 14808 | 4.5e-13 |
| $10^{-8}$ | 85246 | 81.4 | 170570 | 193007 | 97062 | 194086 | 2109 | 4.6 | 6314 | 6403 | 6409 | 5382 | 2.5e-11 |
| $10^{-10}$ | – | – | – | – | – | – | 2318 | 5.0 | 6941 | 6709 | 6715 | 5896 | – |

Table 5: GeoPG-B and APG-B for solving linear regression with elastic net regularization on dataset rcv1

| | APG-B | | | | | | GeoProx-B | | | | | | |
|---|---|---|---|---|---|---|---|---|---|---|---|---|---|
| $\alpha$ | iter | cpu | f-ev | g-ev | p-ev | MVM | f-diff | cpu | f-ev | g-ev | p-ev | MVM | f-diff |
| $\lambda = 1e-03$ | | | | | | | | | | | | | |
| $10^{-2}$ | 18 | 0.1 | 34 | 34 | 20 | 42 | 14 | 0.2 | 39 | 31 | 32 | 43 | 5.5e-14 |
| $10^{-4}$ | 74 | 0.3 | 148 | 141 | 82 | 165 | 95 | 0.7 | 273 | 231 | 232 | 245 | 7.8e-13 |
| $10^{-6}$ | 329 | 1.5 | 678 | 617 | 372 | 735 | 103 | 0.8 | 296 | 265 | 268 | 269 | 6.6e-13 |
| $10^{-8}$ | 908 | 4.2 | 1872 | 1721 | 1033 | 2039 | 133 | 1.0 | 380 | 344 | 345 | 341 | 7.0e-13 |
| $10^{-10}$ | 1277 | 5.9 | 2630 | 2482 | 1454 | 2871 | 116 | 0.9 | 332 | 331 | 332 | 301 | 1.1e-12 |
| $\lambda = 1e-04$ | | | | | | | | | | | | | |
| $10^{-2}$ | 17 | 0.1 | 32 | 31 | 19 | 40 | 17 | 0.1 | 48 | 34 | 35 | 49 | 1.6e-13 |
| $10^{-4}$ | 109 | 0.5 | 226 | 195 | 123 | 243 | 109 | 0.5 | 226 | 195 | 123 | 243 | 1.7e-12 |
| $10^{-6}$ | 723 | 3.2 | 1482 | 1401 | 821 | 1625 | 251 | 1.9 | 743 | 625 | 633 | 634 | 1.3e-11 |
| $10^{-8}$ | 3087 | 13.9 | 6276 | 6426 | 3513 | 6976 | 247 | 2.1 | 723 | 645 | 653 | 626 | 1.4e-11 |
| $10^{-10}$ | 5266 | 23.7 | 10638 | 11244 | 5991 | 11930 | 244 | 1.9 | 711 | 672 | 678 | 624 | 6.0e-12 |
| $\lambda = 1e-05$ | | | | | | | | | | | | | |
| $10^{-2}$ | 16 | 0.1 | 30 | 28 | 18 | 38 | 15 | 0.1 | 42 | 32 | 33 | 45 | 3.1e-13 |
| $10^{-4}$ | 118 | 0.5 | 240 | 220 | 134 | 267 | 125 | 1.0 | 359 | 289 | 294 | 321 | 1.0e-10 |
| $10^{-6}$ | 859 | 3.9 | 1750 | 1595 | 978 | 1941 | 833 | 6.8 | 2470 | 2186 | 2199 | 2105 | 5.7e-10 |
| $10^{-8}$ | 5902 | 26.5 | 11918 | 11933 | 6716 | 13376 | 1179 | 9.6 | 3509 | 3336 | 3348 | 2998 | 1.4e-09 |
| $10^{-10}$ | 33127 | 150.7 | 66438 | 72792 | 37722 | 75353 | 1180 | 9.7 | 3508 | 3540 | 3555 | 2995 | 7.2e-10 |

Table 6: GeoPG-B and APG-B for solving linear regression with elastic net regularization on data set Gisette. Note that neither APG-B nor GeoPG-B converges in $10^5$ iterations when $\mu = 1e - 05$ and $\alpha = 10^{-6}, 10^{-8}, 10^{-10}$.

| $\alpha$ | APG-B | | | | | | GeoPG-B | | | | | | f-diff |
|---|---|---|---|---|---|---|---|---|---|---|---|---|---|
| | iter | cpu | f-ev | g-ev | p-ev | MVM | iter | cpu | f-ev | g-ev | p-ev | MVM | |
| $\mu = 1e - 03$ | | | | | | | | | | | | | |
| $10^{-2}$ | 4026 | 198.1 | 8144 | 7729 | 4583 | 9121 | 4253 | 239.3 | 12593 | 10474 | 10506 | 10758 | 4.8e-14 |
| $10^{-4}$ | 30537 | 1504.2 | 61478 | 61380 | 34786 | 69371 | 6030 | 342.4 | 17939 | 17977 | 18006 | 15411 | 1.6e-13 |
| $10^{-6}$ | – | – | – | – | – | – | 5197 | 294.0 | 15419 | 16126 | 16159 | 13241 | – |
| $10^{-8}$ | – | – | – | – | – | – | 5692 | 322.8 | 16950 | 18851 | 18881 | 14506 | – |
| $10^{-10}$ | – | – | – | – | – | – | 6150 | 353.5 | 18295 | 23420 | 23450 | 15714 | – |
| $\mu = 1e - 04$ | | | | | | | | | | | | | |
| $10^{-2}$ | 6084 | 299.5 | 12288 | 12211 | 6930 | 13801 | 5406 | 304.3 | 16046 | 13623 | 13658 | 13675 | 1.1e-13 |
| $10^{-4}$ | 49467 | 2434.4 | 99880 | 100633 | 56333 | 112194 | 36606 | 2046.7 | 105023 | 112545 | 113414 | 91853 | 1.6e-13 |
| $10^{-6}$ | – | – | – | – | – | – | 20821 | 1179.7 | 62243 | 65886 | 65919 | 53105 | – |
| $10^{-8}$ | – | – | – | – | – | – | 21575 | 1224.1 | 64488 | 71718 | 71753 | 54979 | – |
| $10^{-10}$ | – | – | – | – | – | – | 20328 | 1164.9 | 60730 | 76896 | 76942 | 51908 | – |
| $\mu = 1e - 05$ | | | | | | | | | | | | | |
| $10^{-2}$ | 6570 | 323.9 | 13304 | 13289 | 7483 | 14885 | 4803 | 270.8 | 14228 | 11515 | 11547 | 12164 | 2.7e-13 |
| $10^{-4}$ | 56562 | 2791.0 | 114250 | 115944 | 64396 | 128230 | 38001 | 2153.4 | 113603 | 100036 | 100105 | 96725 | 5.6e-12 |

Table 7: GeoPG-B and APG-B for solving logistic regression with elastic net regularization. $p = 6000, n = 3000$

| $\alpha$ | APG-B | | | | | | GeoPG-B | | | | | | f-diff |
|---|---|---|---|---|---|---|---|---|---|---|---|---|---|
| | iter | cpu | f-ev | g-ev | p-ev | MVM | iter | cpu | f-ev | g-ev | p-ev | MVM | |
| $\mu = 1.00e - 03$ | | | | | | | | | | | | | |
| $10^{-2}$ | 55 | 0.9 | 112 | 96 | 60 | 158 | 46 | 1.3 | 125 | 145 | 146 | 207 | 1.1e-13 |
| $10^{-4}$ | 256 | 4.3 | 536 | 470 | 289 | 761 | 55 | 1.7 | 144 | 194 | 194 | 269 | 5.6e-13 |
| $10^{-6}$ | 509 | 8.7 | 1048 | 972 | 577 | 1551 | 61 | 2.0 | 164 | 218 | 220 | 300 | 1.3e-12 |
| $10^{-8}$ | 573 | 9.5 | 1188 | 1086 | 649 | 1737 | 60 | 1.9 | 161 | 223 | 225 | 305 | 1.4e-12 |
| $10^{-10}$ | 585 | 9.6 | 1208 | 1112 | 663 | 1777 | 59 | 2.1 | 158 | 231 | 233 | 313 | 1.4e-12 |
| $\mu = 1.00e - 04$ | | | | | | | | | | | | | |
| $10^{-2}$ | 51 | 0.7 | 104 | 80 | 55 | 137 | 51 | 1.3 | 141 | 167 | 164 | 236 | 2.5e-13 |
| $10^{-4}$ | 203 | 3.0 | 422 | 336 | 226 | 564 | 118 | 3.2 | 319 | 405 | 396 | 555 | 1.3e-11 |
| $10^{-6}$ | 954 | 14.7 | 1994 | 1662 | 1080 | 2744 | 126 | 3.8 | 335 | 452 | 450 | 614 | 3.1e-11 |
| $10^{-8}$ | 1814 | 28.7 | 3780 | 3311 | 2056 | 5369 | 125 | 3.7 | 336 | 454 | 454 | 614 | 2.6e-11 |
| $10^{-10}$ | 2135 | 33.9 | 4444 | 3952 | 2421 | 6375 | 125 | 4.0 | 336 | 475 | 475 | 635 | 3.2e-11 |
| $\mu = 1.00e - 05$ | | | | | | | | | | | | | |
| $10^{-2}$ | 52 | 0.8 | 102 | 88 | 57 | 147 | 40 | 1.0 | 107 | 129 | 128 | 184 | 2.3e-13 |
| $10^{-4}$ | 141 | 1.9 | 288 | 208 | 154 | 364 | 97 | 2.4 | 257 | 316 | 309 | 438 | 3.3e-11 |
| $10^{-6}$ | 576 | 7.8 | 1246 | 804 | 646 | 1452 | 139 | 4.0 | 350 | 496 | 488 | 669 | 5.6e-11 |
| $10^{-8}$ | 2797 | 38.4 | 6070 | 4014 | 3166 | 7182 | 148 | 4.4 | 372 | 538 | 535 | 723 | 4.0e-10 |
| $10^{-10}$ | 4549 | 63.3 | 9862 | 6703 | 5151 | 11856 | 153 | 4.9 | 392 | 585 | 583 | 776 | 6.2e-10 |

Table 8: GeoPG-B and APG-B for solving logistic regression with elastic net regularization. $p = 3000, n = 6000$

| | APG-B | | | | | | GeoPG-B | | | | | | |
|---|---|---|---|---|---|---|---|---|---|---|---|---|---|
| $\alpha$ | iter | cpu | f-ev | g-ev | p-ev | MVM | iter | cpu | f-ev | g-ev | p-ev | MVM | f-diff |
| $\mu = 1.00e - 03$ | | | | | | | | | | | | | |
| $10^{-2}$ | 58 | 0.9 | 114 | 107 | 63 | 172 | 60 | 1.6 | 169 | 200 | 196 | 279 | 5.1e-14 |
| $10^{-4}$ | 253 | 4.1 | 516 | 466 | 284 | 752 | 110 | 3.5 | 292 | 420 | 412 | 562 | 1.9e-12 |
| $10^{-6}$ | 893 | 15.1 | 1824 | 1757 | 1012 | 2771 | 115 | 4.3 | 305 | 467 | 463 | 615 | 4.1e-12 |
| $10^{-8}$ | 1265 | 21.9 | 2584 | 2543 | 1435 | 3980 | 114 | 4.4 | 302 | 504 | 501 | 649 | 4.9e-12 |
| $10^{-10}$ | 1333 | 22.6 | 2712 | 2691 | 1513 | 4206 | 114 | 4.8 | 302 | 543 | 540 | 688 | 5.0e-12 |
| $\mu = 1.00e - 04$ | | | | | | | | | | | | | |
| $10^{-2}$ | 56 | 0.8 | 112 | 89 | 60 | 151 | 42 | 1.1 | 116 | 133 | 132 | 188 | 1.4e-13 |
| $10^{-4}$ | 159 | 2.2 | 328 | 237 | 174 | 413 | 128 | 3.7 | 340 | 455 | 447 | 616 | 1.7e-11 |
| $10^{-6}$ | 750 | 11.3 | 1560 | 1238 | 845 | 2085 | 157 | 5.2 | 392 | 621 | 614 | 817 | 5.3e-11 |
| $10^{-8}$ | 1927 | 30.3 | 4012 | 3447 | 2182 | 5631 | 158 | 5.8 | 410 | 679 | 674 | 877 | 8.6e-11 |
| $10^{-10}$ | 2364 | 37.5 | 4934 | 4290 | 2677 | 6969 | 164 | 6.6 | 427 | 760 | 753 | 965 | 1.5e-10 |
| $\mu = 1.00e - 05$ | | | | | | | | | | | | | |
| $10^{-2}$ | 54 | 0.8 | 108 | 85 | 58 | 145 | 42 | 1.1 | 110 | 136 | 134 | 191 | 2.9e-13 |
| $10^{-4}$ | 118 | 1.6 | 236 | 177 | 126 | 305 | 81 | 2.1 | 207 | 266 | 263 | 365 | 1.4e-11 |
| $10^{-6}$ | 493 | 6.4 | 1062 | 636 | 551 | 1189 | 153 | 4.9 | 365 | 588 | 580 | 776 | 2.9e-10 |
| $10^{-8}$ | 3492 | 45.0 | 7742 | 4365 | 3949 | 8316 | 163 | 5.8 | 379 | 686 | 677 | 886 | 8.3e-10 |
| $10^{-10}$ | 7655 | 98.4 | 17058 | 9498 | 8666 | 18166 | 169 | 6.8 | 403 | 782 | 775 | 990 | 1.7e-09 |

Table 9: GeoPG-B and APG-B for solving logistic regression with elastic net regularization. $p = 3000, n = 3000$

| | APG-B | | | | | | GeoPG-B | | | | | | |
|---|---|---|---|---|---|---|---|---|---|---|---|---|---|
| $\alpha$ | iter | cpu | f-ev | g-ev | p-ev | MVM | iter | cpu | f-ev | g-ev | p-ev | MVM | f-diff |
| $\mu = 1.00e - 03$ | | | | | | | | | | | | | |
| $10^{-2}$ | 55 | 0.5 | 110 | 99 | 60 | 161 | 53 | 0.8 | 144 | 172 | 171 | 243 | 2.7e-13 |
| $10^{-4}$ | 278 | 2.4 | 566 | 512 | 312 | 826 | 90 | 1.4 | 237 | 325 | 322 | 442 | 2.7e-12 |
| $10^{-6}$ | 845 | 7.1 | 1732 | 1637 | 957 | 2596 | 89 | 1.5 | 234 | 336 | 334 | 452 | 2.6e-12 |
| $10^{-8}$ | 1158 | 9.7 | 2378 | 2283 | 1314 | 3599 | 89 | 1.6 | 234 | 361 | 359 | 477 | 2.6e-12 |
| $10^{-10}$ | 1186 | 9.9 | 2444 | 2340 | 1345 | 3687 | 88 | 1.7 | 231 | 377 | 375 | 492 | 2.8e-12 |
| $\mu = 1.00e - 04$ | | | | | | | | | | | | | |
| $10^{-2}$ | 55 | 0.4 | 108 | 89 | 60 | 151 | 53 | 0.7 | 144 | 172 | 169 | 242 | 3.5e-13 |
| $10^{-4}$ | 172 | 1.3 | 352 | 273 | 191 | 466 | 122 | 1.8 | 327 | 424 | 415 | 579 | 3.2e-11 |
| $10^{-6}$ | 868 | 6.6 | 1834 | 1455 | 980 | 2437 | 145 | 2.3 | 374 | 529 | 523 | 714 | 6.8e-11 |
| $10^{-8}$ | 1985 | 16.0 | 4168 | 3527 | 2248 | 5777 | 144 | 2.5 | 372 | 565 | 563 | 747 | 5.9e-11 |
| $10^{-10}$ | 2475 | 19.9 | 5160 | 4545 | 2807 | 7354 | 143 | 2.7 | 365 | 607 | 605 | 787 | 7.4e-11 |
| $\mu = 1.00e - 05$ | | | | | | | | | | | | | |
| $10^{-2}$ | 55 | 0.4 | 108 | 91 | 59 | 152 | 48 | 0.7 | 129 | 158 | 155 | 224 | 6.7e-13 |
| $10^{-4}$ | 126 | 0.9 | 256 | 185 | 137 | 324 | 126 | 0.9 | 256 | 185 | 137 | 324 | 2.0e-12 |
| $10^{-6}$ | 515 | 3.4 | 1108 | 680 | 576 | 1258 | 146 | 2.2 | 344 | 524 | 517 | 705 | 4.6e-10 |
| $10^{-8}$ | 3196 | 21.0 | 7054 | 4118 | 3615 | 7735 | 154 | 2.5 | 372 | 587 | 586 | 778 | 8.9e-10 |
| $10^{-10}$ | 6434 | 42.7 | 14228 | 8384 | 7284 | 15670 | 152 | 2.8 | 370 | 630 | 629 | 820 | 5.4e-10 |

Table 10: GeoPG-B and APG-B for solving logistic regression with elastic net on dataset a9a

| | APG-B | | | | | | GeoPG-B | | | | | | |
|---|---|---|---|---|---|---|---|---|---|---|---|---|---|
| $\alpha$ | iter | cpu | f-ev | g-ev | p-ev | MVM | iter | cpu | f-ev | g-ev | p-ev | MVM | f-diff |
| $\mu = 1.00e - 03$ | | | | | | | | | | | | | |
| $10^{-2}$ | 99 | 0.3 | 196 | 189 | 111 | 302 | 96 | 0.5 | 280 | 325 | 318 | 450 | 2.9e-15 |
| $10^{-4}$ | 676 | 1.8 | 1380 | 1317 | 766 | 2085 | 676 | 1.8 | 1380 | 1317 | 766 | 2085 | 1.7e-14 |
| $10^{-6}$ | 2696 | 6.8 | 5484 | 5466 | 3065 | 8533 | 187 | 1.0 | 540 | 663 | 683 | 885 | 2.6e-14 |
| $10^{-8}$ | 3911 | 9.8 | 7934 | 8114 | 4445 | 12561 | 188 | 1.0 | 545 | 654 | 678 | 876 | 2.0e-14 |
| $10^{-10}$ | 4324 | 10.9 | 8770 | 9013 | 4917 | 13932 | 200 | 1.1 | 581 | 758 | 783 | 991 | 5.9e-14 |
| $\mu = 1.00e - 04$ | | | | | | | | | | | | | |
| $10^{-2}$ | 96 | 0.2 | 194 | 174 | 106 | 282 | 96 | 0.2 | 194 | 174 | 106 | 282 | 1.1e-14 |
| $10^{-4}$ | 709 | 1.7 | 1440 | 1388 | 805 | 2195 | 756 | 3.8 | 2251 | 2669 | 2577 | 3615 | 8.2e-13 |
| $10^{-6}$ | 5195 | 13.6 | 10488 | 10973 | 5912 | 16887 | 2581 | 13.4 | 7725 | 8995 | 8770 | 12112 | 4.4e-11 |
| $10^{-8}$ | 25300 | 64.8 | 50772 | 56141 | 28793 | 84936 | 716 | 3.7 | 2130 | 2529 | 2583 | 3427 | 9.9e-10 |
| $10^{-10}$ | 42633 | 109.4 | 85446 | 95447 | 48519 | 143968 | 723 | 3.8 | 2151 | 2584 | 2640 | 3497 | 7.9e-11 |
| $\mu = 1.00e - 05$ | | | | | | | | | | | | | |
| $10^{-2}$ | 106 | 0.3 | 210 | 199 | 119 | 320 | 72 | 0.4 | 207 | 258 | 255 | 347 | 1.4e-14 |
| $10^{-4}$ | 770 | 1.9 | 1550 | 1526 | 874 | 2402 | 685 | 3.5 | 2038 | 2448 | 2367 | 3301 | 3.7e-12 |
| $10^{-6}$ | 5842 | 14.7 | 11762 | 12434 | 6648 | 19084 | 3026 | 15.5 | 9061 | 11099 | 10715 | 14750 | 2.1e-11 |
| $10^{-8}$ | 46819 | 119.9 | 93782 | 104946 | 53311 | 158259 | 7784 | 38.8 | 23335 | 26969 | 26326 | 36558 | 1.9e-12 |
| $10^{-10}$ | – | – | – | – | – | – | 1488 | 8.2 | 4447 | 5674 | 5721 | 7567 | – |

Table 11: GeoPG-B and APG-B for solving logistic regression with elastic net on dataset RCV1

| | APG-B | | | | | | GeoPG-B | | | | | | |
|---|---|---|---|---|---|---|---|---|---|---|---|---|---|
| $\alpha$ | iter | cpu | f-ev | g-ev | p-ev | MVM | iter | cpu | f-ev | g-ev | p-ev | MVM | f-diff |
| $\mu = 1e - 03$ | | | | | | | | | | | | | |
| $10^{-2}$ | 15 | 0.1 | 28 | 26 | 17 | 45 | 7 | 0.1 | 21 | 22 | 23 | 36 | 5.0e-14 |
| $10^{-4}$ | 35 | 0.2 | 68 | 61 | 37 | 100 | 30 | 0.3 | 83 | 91 | 93 | 134 | 7.7e-13 |
| $10^{-6}$ | 112 | 0.7 | 224 | 213 | 125 | 340 | 43 | 0.5 | 120 | 136 | 140 | 193 | 1.3e-12 |
| $10^{-8}$ | 196 | 1.2 | 390 | 384 | 220 | 606 | 39 | 0.5 | 110 | 137 | 138 | 191 | 4.3e-12 |
| $10^{-10}$ | 230 | 1.4 | 466 | 444 | 259 | 705 | 39 | 0.5 | 110 | 149 | 150 | 203 | 1.3e-12 |
| $\mu = 1e - 04$ | | | | | | | | | | | | | |
| $10^{-2}$ | 13 | 0.1 | 24 | 22 | 15 | 39 | 11 | 0.1 | 32 | 35 | 36 | 56 | 3.0e-13 |
| $10^{-4}$ | 40 | 0.3 | 80 | 75 | 44 | 121 | 42 | 0.5 | 122 | 136 | 135 | 193 | 3.8e-12 |
| $10^{-6}$ | 178 | 1.0 | 368 | 311 | 200 | 513 | 153 | 1.8 | 431 | 542 | 527 | 738 | 6.2e-11 |
| $10^{-8}$ | 1039 | 6.2 | 2122 | 1941 | 1179 | 3122 | 137 | 1.6 | 384 | 495 | 503 | 673 | 1.9e-11 |
| $10^{-10}$ | 1983 | 11.7 | 4080 | 3724 | 2251 | 5977 | 137 | 1.7 | 379 | 526 | 524 | 705 | 2.5e-12 |
| $\mu = 1e - 05$ | | | | | | | | | | | | | |
| $10^{-2}$ | 13 | 0.1 | 24 | 22 | 15 | 39 | 9 | 0.1 | 26 | 28 | 29 | 45 | 2.3e-13 |
| $10^{-4}$ | 42 | 0.2 | 84 | 71 | 46 | 119 | 39 | 0.4 | 108 | 120 | 122 | 173 | 2.3e-11 |
| $10^{-6}$ | 164 | 0.9 | 338 | 266 | 182 | 450 | 208 | 2.5 | 592 | 747 | 729 | 1012 | 1.4e-09 |
| $10^{-8}$ | 1115 | 6.4 | 2274 | 2013 | 1266 | 3281 | 377 | 4.7 | 1073 | 1436 | 1410 | 1901 | 5.5e-09 |
| $10^{-10}$ | 5569 | 33.5 | 11314 | 11135 | 6334 | 17471 | 486 | 6.2 | 1399 | 1930 | 1897 | 2542 | 6.5e-10 |

Table 13: L-GeoPG-B for solving logistic regression with elastic net regularization on data set Gisette

| | $m = 0$ | | $m = 5$ | | $m = 10$ | | $m = 20$ | | $m = 50$ | | $m = 100$ | |
|---|---|---|---|---|---|---|---|---|---|---|---|---|
| $\alpha$ | iter | cpu | iter | cpu | iter | cpu | iter | cpu | iter | cpu | iter | cpu |
| $\mu = 1e-03$ | | | | | | | | | | | | |
| $10^{-2}$ | 819 | 82.5 | 1310 | 164.1 | 1015 | 125.8 | 902 | 97.0 | 713 | 76.6 | 769 | 93.6 |
| $10^{-4}$ | 2177 | 217.5 | 3656 | 470.8 | 3439 | 417.9 | 3836 | 406.6 | 2399 | 260.1 | 1530 | 185.8 |
| $10^{-6}$ | 2013 | 230.9 | 1606 | 235.9 | 1589 | 221.5 | 1547 | 225.4 | 1344 | 189.6 | 1082 | 168.4 |
| $10^{-8}$ | 1793 | 214.9 | 1622 | 252.7 | 1530 | 224.4 | 1562 | 234.6 | 1363 | 200.8 | 1097 | 172.7 |
| $10^{-10}$ | 1808 | 227.1 | 1599 | 260.8 | 1549 | 245.3 | 1565 | 246.9 | 1369 | 216.9 | 1100 | 180.8 |
| $\mu = 1e-04$ | | | | | | | | | | | | |
| $10^{-2}$ | 961 | 93.7 | 2573 | 312.6 | 2057 | 251.5 | 1487 | 169.6 | 1367 | 137.8 | 1217 | 130.0 |
| $10^{-4}$ | 2146 | 217.2 | 2237 | 312.3 | 2595 | 341.6 | 2621 | 314.0 | 2044 | 242.8 | 1317 | 179.6 |
| $10^{-6}$ | 2243 | 258.2 | 2102 | 307.0 | 2105 | 303.9 | 1979 | 292.2 | 1810 | 272.3 | 1390 | 219.9 |
| $10^{-8}$ | 2226 | 276.7 | 2057 | 329.4 | 2009 | 317.6 | 1951 | 313.6 | 1791 | 288.1 | 1444 | 250.5 |
| $10^{-10}$ | 2203 | 296.2 | 2046 | 361.7 | 2101 | 342.1 | 2002 | 338.3 | 1846 | 307.6 | 1445 | 246.7 |
| $\mu = 1e-05$ | | | | | | | | | | | | |
| $10^{-2}$ | 795 | 79.8 | 3501 | 407.2 | 3022 | 359.3 | 1375 | 166.75 | 1156 | 122.7 | 968 | 106.7 |
| $10^{-4}$ | 1381 | 141.4 | 1461 | 219.2 | 1303 | 179.2 | 1621 | 213.7 | 1198 | 153.5 | 902 | 126.6 |
| $10^{-6}$ | 2928 | 313.0 | 2343 | 352.7 | 2271 | 336.2 | 2179 | 323.8 | 2001 | 297.6 | 1601 | 256.3 |
| $10^{-8}$ | 2946 | 374.2 | 2401 | 380.0 | 2349 | 380.5 | 2254 | 360.8 | 2099 | 345.6 | 1804 | 303.9 |
| $10^{-10}$ | 2776 | 436.7 | 2503 | 432.4 | 2363 | 414.3 | 2350 | 409.4 | 2093 | 365.7 | 1826 | 320.2 |

Table 12: GeoPG-B and APG-B for solving logistic regression with elastic net on dataset Gisette

| | APG-B | | | | | | GeoPG-B | | | | | | |
|---|---|---|---|---|---|---|---|---|---|---|---|---|---|
| $\alpha$ | iter | cpu | f-ev | g-ev | p-ev | MVM | iter | cpu | f-ev | g-ev | p-ev | MVM | f-diff |
| $\mu = 1e-03$ | | | | | | | | | | | | | |
| $10^{-2}$ | 630 | 40.4 | 1267 | 1176 | 717 | 1895 | 819 | 82.5 | 2298 | 2867 | 2790 | 3903 | 2.8e-14 |
| $10^{-4}$ | 2445 | 156.0 | 4923 | 4511 | 2784 | 7297 | 2177 | 217.5 | 6197 | 7710 | 7477 | 10406 | 3.9e-13 |
| $10^{-6}$ | 13950 | 915.2 | 28209 | 28106 | 15889 | 43997 | 2013 | 230.9 | 5654 | 7676 | 7737 | 10200 | 2.0e-12 |
| $10^{-8}$ | 64288 | 4397.1 | 129271 | 140483 | 73191 | 213676 | 1793 | 214.9 | 5033 | 7146 | 7188 | 9371 | 4.4e-14 |
| $10^{-10}$ | – | – | – | – | – | – | 1808 | 227.1 | 5079 | 7532 | 7559 | 9783 | – |
| $\mu = 1e-04$ | | | | | | | | | | | | | |
| $10^{-2}$ | 913 | 57.7 | 1845 | 1744 | 1041 | 2787 | 961 | 93.7 | 2740 | 3335 | 3237 | 4553 | 3.5e-13 |
| $10^{-4}$ | 1889 | 113.1 | 3811 | 3246 | 2150 | 5398 | 913 | 57.7 | 1845 | 1744 | 1041 | 2787 | 3.9e-12 |
| $10^{-6}$ | 10206 | 614.4 | 20687 | 17730 | 11632 | 29364 | 2243 | 258.2 | 6044 | 8768 | 8763 | 11486 | 3.0e-11 |
| $10^{-8}$ | 53272 | 3405.7 | 107397 | 103641 | 60702 | 164345 | 2226 | 276.7 | 6001 | 9318 | 9300 | 12002 | 2.8e-11 |
| $10^{-10}$ | – | – | – | – | – | – | 2203 | 296.2 | 5926 | 9809 | 9812 | 12488 | – |
| $\mu = 1e-05$ | | | | | | | | | | | | | |
| $10^{-2}$ | 975 | 63.2 | 1981 | 1882 | 1110 | 2994 | 795 | 79.8 | 2242 | 2738 | 2662 | 3747 | 6.5e-13 |
| $10^{-4}$ | 1485 | 91.1 | 3019 | 2632 | 1687 | 4321 | 1381 | 141.4 | 3760 | 4943 | 4829 | 6686 | 6.8e-12 |
| $10^{-6}$ | 4642 | 265.8 | 9439 | 7240 | 5286 | 12528 | 2928 | 313.0 | 7964 | 10940 | 10698 | 14554 | 1.5e-11 |
| $10^{-8}$ | 29242 | 1681.8 | 59811 | 46411 | 33325 | 79738 | 2946 | 374.2 | 7789 | 12617 | 12543 | 16128 | 5.5e-10 |
| $10^{-10}$ | – | – | – | – | – | – | 2776 | 436.6 | 7359 | 13607 | 13563 | 16936 | – |