[Reviews · NeurIPS 2017]

Reviewer 1



Summary: The paper extends the recent work of Bubeck et al. on the geometric gradient descent method, in order to handle non-smooth (but “nice”) objectives. Similar work can be found in [10]; however, no optimal rates are obtained in that case (in the sense of having a square root condition number in the retraction factor). The core ideas and motivation origin from the proximity operators used in non-smooth optimization in machine learning and signal processing (see e.g., lasso). Quality: The paper is of good technical quality - for clarity see below. The results could be considered “expected” since similar results (from smooth to non-smooth convex objectives with proximal operators) have been proved in numerous papers the past decade; especially under convexity assumptions. Technical correctness: I believe that the results I read in the main are correct. Impact: The paper might have a good impact on the direction of demystifying Nesterov’s accelerated routines like FISTA (by proposing alternatives). I can see several ramifications based on this work, so I would say that the paper has at least some importance. Clarity: My main concern is the balance of being succinct and being thorough. The authors tend towards the former; however, several parts of the paper are no clear (especially the technical ones). Also, the whole idea of this line of work is to provide a better understanding of accelerated methods; the paper does not provide much intuition on this front (update: there is a paragraph in the sup. material - maybe the authors should consider moving that part in the main text). Comments: 1. Neither here, nor in the original paper of Bubeck it is clear how we get some basic arguments: E.g., I “wasted” a lot of time to understand how we get to (2.2) by setting y = x^* in (2.1). Given the restricted review time, things need to be crystal clear, even if they are not in the original paper, based on which this work is conducted. Maybe this can be clarified by the rest of the reviewers. Assuming y = x++, this gives the bound x* \in B(x++, 2/\alpha^2 * || \nabla f(x) ||_2^2 - 2/\alpha (f(x) - f(x++))), but f(x*) \leq f(x++), so we cannot naively substitute it there. Probably I’m missing something fundamental. Overall: There is nothing strongly negative about the paper - I would say, apart from some clarity issues and given that proofs work through, it is a “weak accept”.

Reviewer 2



The paper proposes an extension of the geometric descent method of Bubeck, Lee and Singh proposed [1]. The extension is to the case of composite optimization objectives, something which was already achieved by Drusvyatskiy, Fazel and Roy [10] for a (slightly suboptimal) variant of the algorithm. The theory part of the paper finally provides a useful discussion of backtracking (which is an ok but not earth-shaking contribution), and ends with a limited memory discussion which however seems not to give much novel on top of [9,10]. The experiments are comparing to Nesterov's accelerated method, and show in my view very promising results for the geometric method. The experiments are performed on very relevant machine learning models and real datasets. For reproducibility: What was used to stop algorithm 1 + 2 (the internal subroutines? See also next comment). Summary: I find this topic highly interesting, and it could prove fruitful in view of the huge popularity but limited theoretical understanding of accelerated methods and momentum in deep learning and other applications. My main concern is if the paper here makes enough new contributions on top of the already existing work of [1,9,10], some of which have already covered the composite case. The paper though is nicely written and I find the experiments convincing. Comments: For satisfying the necessary conditions in each step, the authors proposed the Brent-Dekker method and semi-smooth Newton for two univariate subproblems. Please comment more clearly that those subproblems are univariate, and give more details why/when these two algorithms are suitable, along with discussing the complexity and impact for overall complexity, in contrast to e.g. binary search. Crucially: What accuracy is needed for the convergence to hold? (Exact root finding will be intractable in both Algo 1 and 2 I assume). == Update after author feedback == Thanks for answering my question; keeping my assessment for now